# Simulation Models of Skidder Conventional and Hybrid Drive

**Juraj Karlušić** [1], **Mihael Cipek** [1], **Danijel Pavković** [1], **Juraj Benić** [1], **Željko Šitum** [1], **Zdravko Pandur** [2] and **Marijan Šušnjar** [2,*]

[1]  Department of Robotics and Production System Automation, Faculty of Mechanical Engineering and Naval Architecture, University of Zagreb, 10000 Zagreb, Croatia; juraj.karlusic@fsb.hr (J.K.); mihael.cipek@fsb.hr (M.C.); danijel.pavkovic@fsb.hr (D.P.); juraj.benic@fsb.hr (J.B.); zeljko.situm@fsb.hr (Ž.Š.)

[2]  Forest Engineering Institute, Faculty of Forestry, University of Zagreb, 10000 Zagreb, Croatia; zpandur@sumfak.hr

*  Correspondence: msusnjar@sumfak.hr; Tel.: +385-99-310-2862

**Abstract:** The paper presents a hypothetical conversion of a conventional cable skidder powertrain to its hybrid version. Simulations of skidder operation were made for two existing forest paths, based on the technical characteristics of the engine, transmission system and the characteristics of the winch. Fuel and time consumption were calculated per working cycle considering the operating conditions (slope, load mass). The model was then converted to a hybrid version by adding a battery energy storage system in parallel with the electrical power generator and by employing an energy management control strategy. The dimensions of the battery and the power generator were chosen based on the characteristics of the existing winch with the aim of completely taking over its operation. The management strategy was selected using the specific fuel consumption map. All simulations were repeated for the hybrid drive under the same operating conditions. The results show that fuel savings of around 13% can be achieved with the selected hybrid drive and steering strategy.

**Keywords:** skidder; simulation; hybrid drive; fuel consumption

## 1. Introduction

Accelerated global warming encourages science and industry to research and make changes in technology in favor of Greenhouse gases (GHG) reduction. The public is already familiar with the electrification and hybridization of personal automobiles [1,2]; however, the same approaches can be applied to non-road mobile machinery (NRMM). These include mobile machines and transportable industrial equipment, which are not intended for transporting goods or passengers on roads and are also fitted with an internal combustion engine (ICE) as the primary on-board energy source [3]. Their common feature is that all are intended for intensive labor [4] and are often operated on a daily basis with eight-hour or even longer work shifts [5]. According to [6], agriculture and forestry takes six percent of (GHG) shares in EU-27 countries. During forest harvesting operations, fuel consumption contributes to a large portion of the overall environmental impacts compared to other resources consumed (tires, hydraulic/motor oil, spare parts, etc.), making it the most consumed resource in these operations, while the emission factors associated with the combined provision and consumption of diesel and gasoline fuels were among the highest in comparison with other inventory categories [7]. In the overall energy audit of mechanized wood harvesting systems in Ireland, fuel consumption was the most significant item (82%), followed by oils and other lubricants (7%) and machine repairs and replacement (11%) [8,9].

Lately, many manufacturers are presenting their variants of hybrid NRMM vehicles used in agriculture and forestry, some of which are "cut to length" tractors of a hydraulic [10] or an electric hybrid [11] variant. Hybridization yields the following three main advantages compared to traditional powertrains: lower fuel consumption, better driving performance and reduction in harmful gases and particles [12]. Good knowledge of operating modes and vehicle purpose is crucial when selecting the appropriate hybrid power-train topology and designing the overall hybrid vehicle system. Similarly, when retrofitting the existing vehicle with the hybrid propulsion system, it is also very important that the retrofitting requirements do not result in extensive redesign and costly interventions within the existing on-board power-train and vehicle chassis.

Having this in mind, the main hypotheses of the presented work are that: (i) it would be possible to convert a conventional 84 kW diesel-powered skidder currently found in the national forestry company fleet, to a battery hybrid counterpart; (ii) and that it would result in notable fuel and greenhouse gas emissions for the several operating scenarios located within the Lika region (Republic of Croatia).

The paper is organized as follows. Section 2 describes the model of a typical diesel-powered skidder, with parameters obtained from the publicly-available data. Overall energy requirements for different driving and operating patterns are calculated in Section 3. These results are used in Section 4 for the sizing of the main generator and battery energy storage system of the battery-hybrid skidder counterpart, and subsequent simulation analysis, while the comparative techno-economic aspects of the proposed upgrade are given in Section 5. The concluding remarks are given in Section 6.

## 2. Skidder Simulation Models

The EcoTrac 120 V skidder, used for pulling logs out of a forest, is considered in this work. It is manufactured by the Hittner company located in the town of Bjelovar, Croatia [13]. According to the data from [14], more than a hundred EcoTrac 120 V units are used by different forestry estates in Croatia. In this section, simulation models of conventional and hybrid skidders are given.

### 2.1. Conventional Skidder Model

What separates skidders from other off-road machinery is that they have a double (or single) drum winch used for pulling logs and a rear anchor-protective board for the purpose of protecting the rear side of the skidder and additional stability during working with the winch. Currently, the aforementioned winch and other hydraulic actuators are driven by a hydraulic pump. Hence, the conventional skidder model consists of the skidder driver (human operator) sub-model, static maps for the internal combustion engine and gearbox torque vs. speed relationships and efficiency, engine fuel consumption map, winch system and the sub-model of skidder longitudinal dynamics.

#### 2.1.1. Engine and Gearbox Model

The main power source is the 6.5 liter six-cylinder in-line (straight-six), air-cooled, four stroke diesel engine manufactured by Deutz AG, type F6L-914. It is a naturally-aspirated engine with maximum power rating $P_e$ of 84 kW at 2300 min$^{-1}$ and maximum engine torque $T_e$ of 400 Nm at 1500 min$^{-1}$ [15]. The maximum torque characteristic for this particular engine is represented by the blue curve in Figure 1a extracted from the manufacturer's documentation [16]. In order to determine the specific torque values for different engine throttle positions (other curves in Figure 1a), torque vs. throttle points from a similar engine are adopted for the engine in question based on data in [17], because these data were not provided by the manufacturer. Linear interpolation has been used to calculate engine torques for all possible throttle positions between individual curves shown in Figure 1a. The optimum engine operating range is from 1300 to 1800 min$^{-1}$ at about 360 Nm, with specific fuel consumption within this operating range being between 200 and 214 g/kWh. Instantaneous fuel consumption is calculated by multiplying the specific consumption with the instantaneous engine power, and it naturally increases with the increase in the engine rotational speed and load torque. Data provided by the manufacturer in [16] contains only the specific fuel consumption value at the

rated engine power; therefore, the fuel consumption map shown in Figure 1a is created by adapting the fuel consumption map of a similar engine from [18].

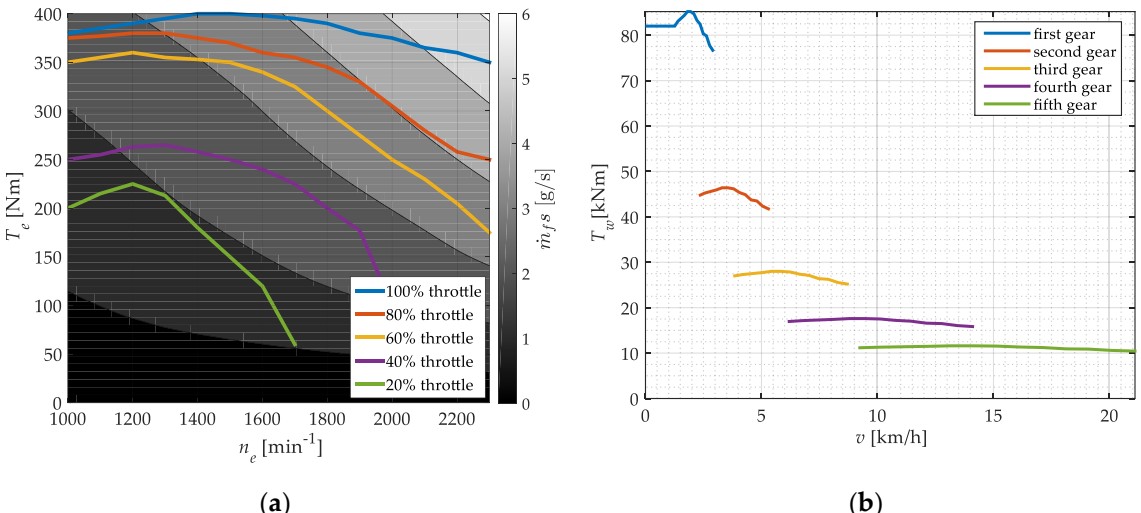

**Figure 1.** Fuel consumption map and torque curves dependent on throttle position (**a**), and output torque for selected gear and vehicle speed (**b**).

Engine mechanical power is transferred through a ten-speed manual transmission and transfer case to all four wheels. The transmission system consists of five low-ratio and five high-ratio gears [15]. Low ratio gears are particularly interesting for this study, because they are used for operating in forest areas and will be considered for the analysis in the remainder of this paper. The total transmission ratios for each low-ratio gear setting, transfer case and differentials with top speeds for particular gears are shown in Table 1, while the power-train torque dependence with respect to the vehicle velocity and selected low-speed gear ratio is shown in Figure 1b. Note that the torque curves in Figure 1b do not overlap because of large differences between individual gear ratios. Note also that the horizontal portion of the low-velocity first gear characteristic (blue curve) in Figure 1b is a simplification where the clutch slips and, thus, the transmitted torque is practically constant.

**Table 1.** Gear ratios and top speed [12].

| Gear | Total Gear Ratio (-) | Top Speed at 2300 min$^{-1}$ (km/h) |
|:---:|:---:|:---:|
| 1 | 213 | 2.9 |
| 2 | 116 | 5.3 |
| 3 | 70 | 8.7 |
| 4 | 44 | 14.1 |
| 5 | 29 | 21.1 |

### 2.1.2. Winch

The double drum winch has a maximum pulling distance of 70 m and cable winding speed of 1.26 m/s [19]. Upon arrival at the extraction site, workers link logs with the winch cable, and the skidder subsequently pulls logs towards the protective board. During these operations, the skidder engine is in the idle speed control regime, while the skidder itself is stationary (with brakes engaged). For the purpose of analysis, the loaded winch is modelled as a cable-hung mass on a slope with friction referred to the pulley side of the winch. Power is transferred from the hydraulic pump to the winch drum via the worm drive with the reduction ratio of 20.5 and characterized by a constant-valued mechanical power transfer efficiency of 0.86 [19].

### 2.1.3. Longitudinal Dynamics

Forces acting upon the skidder during skidding (driving and pulling weight) are shown in Figure 2, and are defined according to [20], where log is defined as a rigid object with a point load. Based on this description, the total force acting upon the skidder is calculated from the force equilibrium as follows:

$$F_l = Q(1-k)\left[\mu_p \cos(\alpha) - \sin(\alpha)\right] + (G + kQ)[f \cos(\alpha) - \sin(\alpha)], \tag{1}$$

where $F_l$ is total force, $Q$ is log weight in N, $G$ is skidder weight in N, $\alpha$ is terrain slope in rad, and where $f$, $k$ and $\mu_p$ are rolling resistance, load mass distribution and skidding resistance coefficients, respectively. Rolling resistance $f$ and skidding resistance $\mu_p$ are highly dependent on the varying state of terrain (e.g., muddy, dry or wet ground, soil of rocky terrain). The terrain slope sign defines whether uphill or downhill driving is taking place, and its value is limited according to the maximum values obtained during conducted measurements in the field [21]. Within the scope of work presented herein, these factor values will be varied around median values estimated from the field data ($f = 0.12$, $k = 0.48$, $\mu_p = 0.51$) [22]. The distribution of forces acting upon the skidder and the load is illustrated in Figure 2.

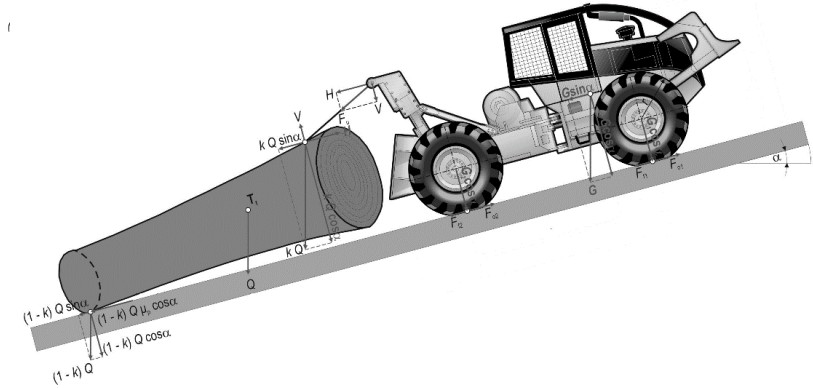

**Figure 2.** Forces acting upon the skidder and load during skidding [20].

### 2.1.4. Gear Selection

A straightforward virtual driver logic is used in order to change gears depending on engine rpm. The same logic is also subsequently used within the hybrid skidder power-train. In particular, when the engine speed crosses 1900 $\text{min}^{-1}$, the virtual driver shifts into the higher gear, whereas if the engine speed falls below 1100 $\text{min}^{-1}$, transmission is shifted into the nearest lower gear. In every time instant, wheel torque in the higher gear is compared with the current load and if the load exceeds the threshold value, the driver then shifts to a higher gear when passing 1900 $\text{min}^{-1}$. In the case where the engine rpm passes 1900 $\text{min}^{-1}$ but wheel torque in the higher gear becomes lower than the current wheel load, the driver does not upshift and the velocity limiter activates (thus emulating the real driver who wants to prevent wheel slipping). Limit velocity is set at a predefined value above 2000 $\text{min}^{-1}$, depending on the selected gear ratio. An additional logic condition is introduced, so that when rpm is higher than 2000 $\text{min}^{-1}$, it activates when the load becomes smaller than the wheel torque in the higher gear.

### 2.1.5. Overall Conventional Skidder Model

A simulation model of the conventional power-train skidder is shown in Figure 3. The virtual driver, who is labelled with dotted lines, provides the throttle valve target position signal, which is forwarded towards the engine (represented by the torque map block), and the braking signal, which is multiplied by the breaking potential and forwarded towards the brakes. The engine torque block contains static maps, which are used to calculate engine torque and fuel consumption from the commanded throttle valve position. In order to calculate the vehicle acceleration, the sum of all forces acting upon the skidder (i.e., wheel force $F_w$, load $F_l$ and breaking force $F_b$) is calculated (summation

point in Figure 3). Since the skidder is represented as point mass, its overall mass consists of the skidder mass itself $m_s$ and the log mass $m_t$.

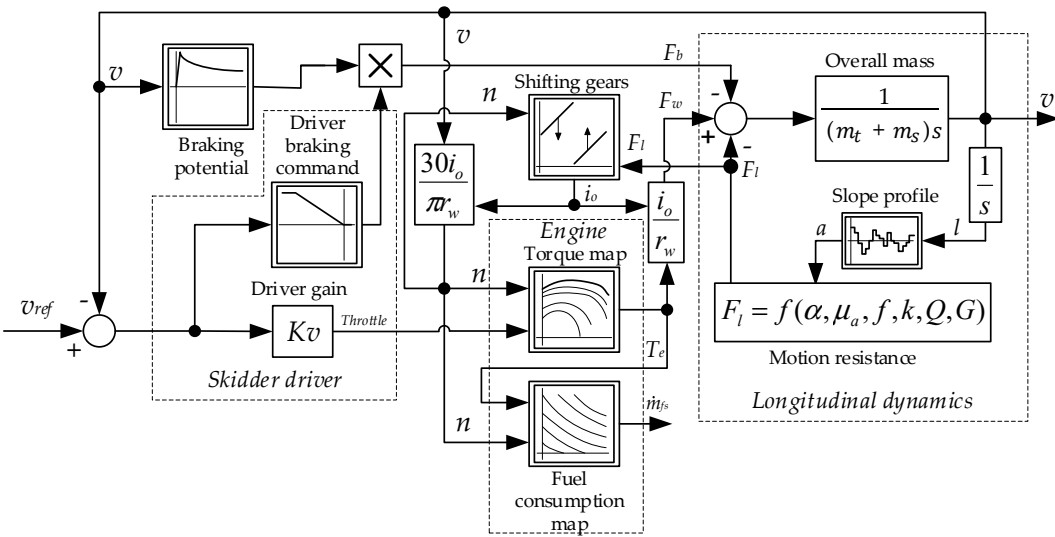

**Figure 3.** Block diagram of a conventional skidder power-train and driveline control strategy.

For the sake of simplicity, a virtual driver is implemented as a proportional speed controller with proportional gain $Kv$, thus resulting in acceptable levels of accuracy of speed target following. The virtual driver acts upon the velocity difference between the reference velocity $v_{ref}$ and skidder velocity $v$, and produces the control outputs (i.e., throttle valve and brake pedal position). The heuristic value of driver proportional gain $Kv = 250$ is chosen herein, which yields a fast response and small skidder velocity target following error. Braking action is activated when the velocity error becomes negative.

## 2.2. Hybrid Powertrain

The principal requirement of hybridization is maintaining the performance of the hybrid drive equal to the conventional drive, while requiring minimal modifications to the overall power-train. Performance gains such as improved acceleration, driving comfort, etc. are not crucial for this kind of vehicle. Hence, the introduction of hybrid power-train in this type of vehicle is primarily motivated by the reduction in its fuel consumption and GHG emissions. Fuel savings can be achieved by implementing a so-called start–stop functionality that turns off the engine when the vehicle is stationary and, in that case, the electric motor (electromotor) takes over the driving of hydraulic pumps. Operating regimes with peak engine torque, i.e., when hard accelerating and going uphill are encountered, can also be covered with the additional torque from the electric motor, thus further reducing the fuel consumption. It is important for a hybrid system not to demand substantial modifications to the existing vehicle, so a P2 parallel hybrid configuration is proposed herein. It satisfies all of the listed demands and does not require major modifications of the conventional power-train for its installation [23,24]. The proposed hybrid power-train configuration is shown in Figure 4.

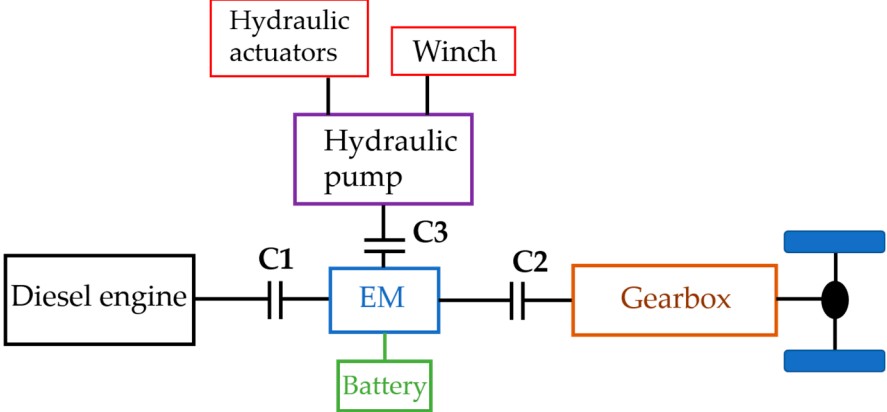

**Figure 4.** Proposed hybrid power-train configuration.

Listed requirements can be divided into the so-called work regimes. Work modes of the hybrid drive train depending on work regime are shown in Table 2.

**Table 2.** Operating modes of hybrid vehicle.

| Work Regime | Diesel Engine | Electromotor | Clutch 1 | Clutch 2 | Clutch 3 |
| --- | --- | --- | --- | --- | --- |
| In Optimal | On | Off | Engaged | Engaged | Disengaged |
| Low Power | On | Generator | Engaged | Engaged | Disengaged |
| High Power | On | Motor | Engaged | Engaged | Disengaged |
| Regenerative Braking | On | Generator | Disengaged | Engaged | Disengaged |
| Winching | Off | Motor | Disengaged | Disengaged | Engaged |

Main requirement for getting into the specified operating mode is determined by the current (actual) battery state of charge (SoC). Although in some cases torque from electromotor would be enough for vehicle propulsion, because of unpredictable driving conditions and sudden changes in power demand, the diesel engine will always be turned on during driving.

### 2.2.1. Electromotor Selection

An electromotor is used to propel the winch hydraulic system as well as any remaining actuators. Since the winch drive is the largest power consumer during stationary skidder operation, the battery and electromotor are chosen to match its loading characteristics. In the particular application, the largest pulling force is 66 kN and the maximum pulling velocity is 1.26 m/s, which gives the maximum power requirement of 83.16 kW. A synchronous AF 130 motor with nominal torque of 145 Nm and nominal power of 64 kW is chosen [25]. Its peak output power of 100 kW can be delivered for up to 60 s, which is considered satisfactory for the particular application. Torque and power characteristics of the AF 130 motor are shown in Figure 5a,b and listed in Table 3.

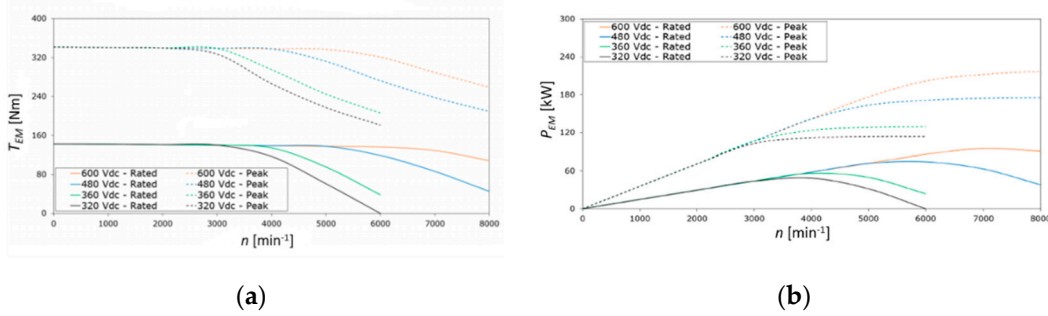

(**a**)  (**b**)

**Figure 5.** AF 130 electromotor torque curve (**a**), and power curve (**b**) [25].

**Table 3.** AF130 electromotor data [25].

| Parameters | *AF* **130** |
|---|---|
| Nominal Torque $N_{EM}$ [Nm] | 145 |
| Nominal Power $P_{EM}$ [kW] | 64 |
| Peak Power—up to 60 s [kW] | 100 |
| Radius-Length | 300–110 |
| Mass $m_{EM}$ [kg] | 30.5 |

### 2.2.2. Battery

For the purpose of battery energy storage system design, Li-Ion cells are chosen due to their availability and high gravimetric energy density. The number of cells required (250 cells in particular) is determined so the battery is capable of delivering enough power at maximum winch load (83 kW). They are divided in two parallel blocks, each with 125 cells in series, which together can deliver 100 kW. The parameters of a single cell [26] and the proposed battery are given in Table 4. Equivalent circuit, voltage characteristics and internal battery resistance are shown in Figure 6.

**Table 4.** Cell and battery parameters.

| | Number N | Energy, E (kWh) | Capacity Q (Ah) | Power, $P_{max}$ (kW) | Mass, $m_{bat}$ (kg) |
|---|---|---|---|---|---|
| Cell | 1 | 0.06 | 15.9 | 0.4 | 0.63 |
| Battery | 250 | 15 | 31.8 | 100 | 157.2 |

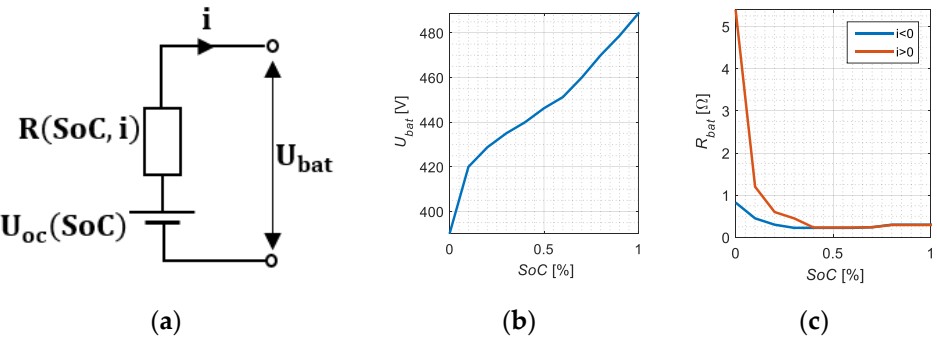

(**a**)  (**b**)  (**c**)

**Figure 6.** Battery equivalent circuit (**a**), battery voltage vs. state-of-charge (SoC) dependence (**b**), internal resistance vs. SoC and current direction (**c**).

The battery model is derived from the equivalent circuit shown in Figure 6a, according to the well-known relationship [27]:

$$\frac{dSoC}{dt} = \frac{\sqrt{U_o^2(SoC) - 4R(SoC,i)\cdot P_{bat}} - U_o(SoC)}{2Q_{maks}\cdot R(SoC,i)},\tag{2}$$

where *SoC* is the battery State-of-Charge, *R* internal resistance, $U_o$ battery voltage, $Q_{max}$ battery capacity and $P_{bat}$ battery power flow.

Total weight of the proposed battery and electromotor is 188 kg, which represents an increase of only 2.6% with respect to total skidder mass. Although other additional hybrid system components such as cooling, DC/AC inverters and other supporting parts must be included in the overall design, the total mass increase should not perceptibly alter the driving characteristics of the modified vehicle.

### 2.2.3. Hybrid Control Strategy

Hybrid drive control arranges power between the diesel and electric motors. A hybrid control unit is added within the vehicle model, in particular, between the driver and the diesel engine and the electric motor (Figure 7). A control strategy is implemented with constant parameters (non-adaptive) regardless of load size and trip length.

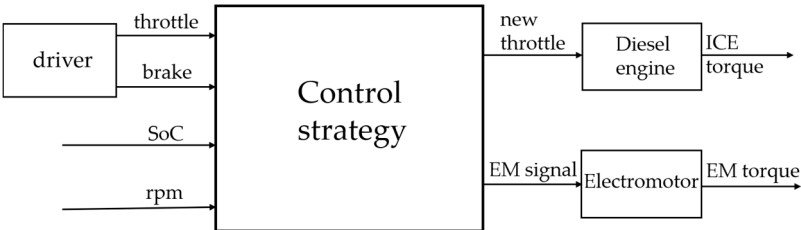

**Figure 7.** Hybrid control module.

The throttle and brake pedal, from the driver, first go to the control unit, which, depending on its input states, provides the control action to both the ICE and the electric motor (EM). Four different operating modes are possible: only ICE, ICE and EM, ICE that recharges the battery, and regenerative braking mode.

When SoC drops below the limit value set at 60%, the diesel engine recharges the battery during normal driving operation. The brake pedal position is checked in every time step, and, if pushed, it cancels the battery recharging by means of the diesel engine. If both conditions are satisfied, control unit checks whether the requested torque $T_{rq}$ from driver is lower than the optimal torque $T_{opt}$. The most efficient area is between 1300 and 1800 min$^{-1}$. If all four conditions are satisfied, the control unit gives a request for larger throttle command to the diesel engine control unit so that it can produce enough torque for driving and recharging. The above logic is depicted by the pseudo code:

IF *SoC* < 60% AND *brake* = 0 AND *rpm* > 1300 && < 1800 AND $T_{rq} < T_{opt}$

    *new throttle* = torque map (*rpm*, $T_{opt}$)
        *EM signal* = $T_{opt} - T_{rq}$

There are three different possibilities for EM torque boosting. If the battery SoC is over 40%, it is possible to turn the EM on to assist the diesel engine. More precisely, if the wanted engine torque is higher than the optimal torque, the control unit sends a new throttle signal to the engine, which then outputs torque equal to the optimal value, while the remaining load is covered by the EM torque.

IF SoC > 40% AND *Trq* > *Topt*

$$EM\ signal = Topt - Trq$$

Only a small work domain is covered that way, so two more conditions are added:

- EM helps with full torque value if the engine operating point crosses 1850 $\text{min}^{-1}$ and the throttle is pushed. On driving routes with frequent inclines and smaller loads, battery saturation is possible. To overcome that problem, a third requirement is added:

  IF rpm > 1850 AND throttle > 0 AND SoC > 40%
  　　*EM signal = 145*

- If battery SoC exceeds 85%, EM turns on and helps with torque regardless of engine operating point:

  IF *SoC* > 85% AND $T_{rq}$ > 200
  　　*EM signal = 145*

When braking occurs, a signal from brake pedal is registered, and if SoC is below 90%, an additional command is given to the electromotor (EM) to start operating as a generator and, depending on the brake pedal position, to provide additional braking force. Total braking force from EM depends on the current gear ratio of the skidder transmission. If necessary, any excess of the braking force over the maximum EM braking torque is achieved with brake discs. The control logic for regenerative braking is below, where *Pk* is brake amplification to convert the brake pedal command to braking force.

IF *SoC* < 90% AND *brake* > 0

　　*EM signal = brake \* Pk \* gear ratio*

## 3. Skidder Daily Working Cycle

A skidder's work day is divided into driving cycles, which consist of driving and winching operations. Skidders are typically used over shorter distances, where they are most efficient [14]. Simulations are based on the productive time spent winching and skidding, which means that skidders work an ideal full shift every work day, without any breaks between operations. Both conventional and hybrid skidders will have the same working times.

### 3.1. Driving Routes

In order to calculate the fuel consumption and total operating time, two different driving routes are defined by using publicly available data from Internet services (Croatian forestry company "Hrvatske šume" and "GPS Visualizer"). All routes are located near the town of Otočac in the Lika region of Croatia.

The skidder's operation starts in the warehouse, marked with the "home" icon on the map (Figure 8b). It then drives on the route marked 0 to the landing site marked with the "truck" icon. Within one work day, the skidder passes route 0 only twice (going from and to the warehouse); therefore, this route is omitted from the fuel consumption calculations. Throughout the work day, skidder drives over routes 1 and 2 marked with red and green lines, i.e., from the landing site to the felling site and back. In the felling site, the skidder hooks logs and transports them to the landing site.

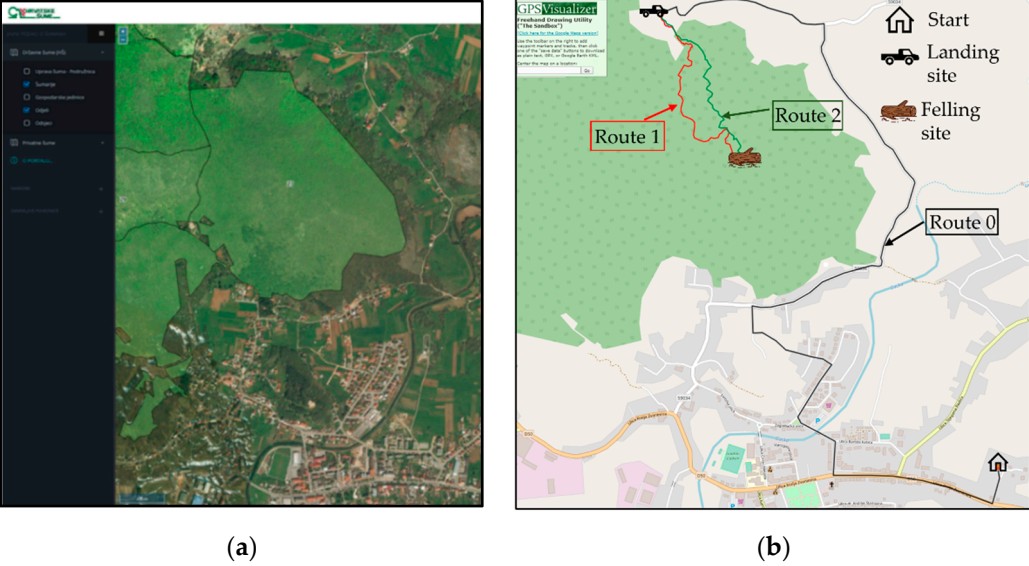

<p align="center">(<b>a</b>)         (<b>b</b>)</p>

**Figure 8.** Driving routes for the skidder's work day: satellite image (**a**) and map layout (**b**).

Driving routes are produced by using the internet service "GPS Visualizer", and they consist of interconnected points in 3D space forming polygons. The geographic latitude, longitude and elevation of each point are known, so it is necessary to determine the distance between each point, for which the Haversine formula is used:

$$l_{t,\,i} \;=\; 2R\cdot\sin^{-1}\!\left(\sqrt{\sin^2\!\left(\frac{s_{i+1}-s_i}{2}\right)\;+\;\cos(s_i)\cdot\cos(s_{i+1})\cdot\sin^2\!\left(\frac{d_{s+1}-d_s}{2}\right)}\,\right), \tag{3}$$

where $l_{t,i}$ is the length between two points, $R$ is the Earth's radius of 6371 km, $s_i$ is the geographical latitude and $d_i$ is the geographical longitude of the $i$-th point. Passing through every point in the loop, the distance between every point and the starting one is calculated. Knowing the distance between points, the angle of each polygon is calculated according to the following expression:

$$\alpha_i \;=\; \tan^{-1}\!\left(\frac{a_{i+1}-a_i}{l_{i+1}-l_i}\right), \tag{4}$$

where $\alpha_i$ is the angle of the $i$-th polygon, and $\alpha_i$ the elevation of the $i$-th point. Elevation profiles and terrain slope for two routes are shown in Figure 9. Even though routes are presented only in one direction, the skidder drives over every one of them in both directions, so these are discriminated as uphill and downhill routes.

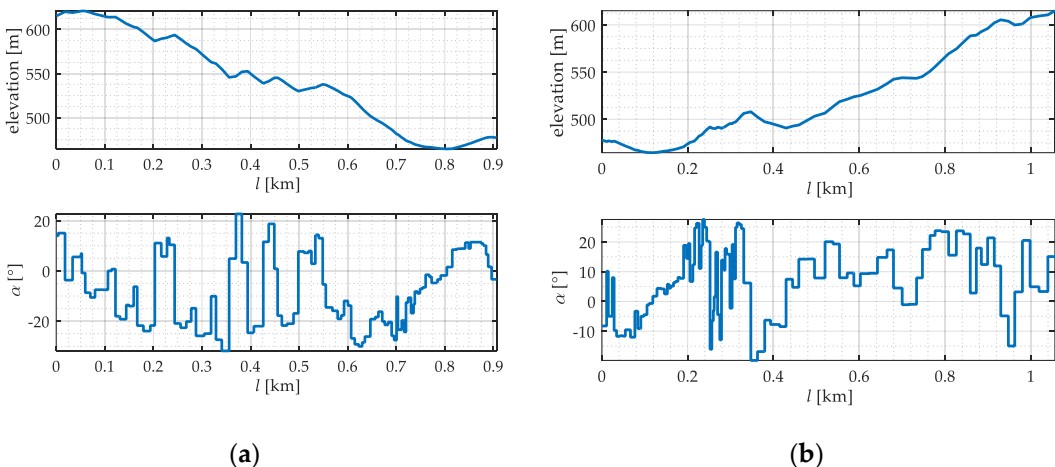

**Figure 9.** Elevation and terrain slope (**a**) of route 1, and (**b**) of route 2.

A total of 16 driving scenarios that consist of two driving routes with 8 different sets of load mass and terrain parameters are defined and shown in Table 5. They will be used in the simulation for both the conventional and hybrid vehicles and their results shown in Section 4.

**Table 5.** Driving scenarios.

| Driving Scenario | Route-dir.–log mass $m_t$ [t] | Skidding Length l [km] | Rolling Resistance f | Load Mass Distribution k | Skidding Resistance Coefficient $\mu_p$ |
|---|---|---|---|---|---|
| 1.1 | 1-uphill-0 | 1.06 | 0.17 | 0 | 0 |
| 1.2 | 1-downhill-4.5 | 1.06 | 0.17 | 0.36 | 0.46 |
| 1.3 | 1-uphill-0 | 1.06 | 0.15 | 0 | 0 |
| 1.4 | 1-downhill-6 | 1.06 | 0.15 | 0.56 | 0.4 |
| 1.5 | 1-uphill-0 | 1.06 | 0.13 | 0 | 0 |
| 1.6 | 1-downhill-1.3 | 1.06 | 0.10 | 0.45 | 0.4 |
| 1.7 | 1-uphill-0 | 1.06 | 0.10 | 0 | 0 |
| 1.8 | 1-downhill-3.6 | 1.06 | 0.15 | 0.43 | 0.6 |
| 2.1 | 2-uphill-0 | 0.907 | 0.15 | 0 | 0 |
| 2.2 | 2-downhill-3 | 0.907 | 0.15 | 0.6 | 0.5 |
| 2.3 | 2-uphill-0 | 0.907 | 0.17 | 0 | 0 |
| 2.4 | 2-downhill-6 | 0.907 | 0.18 | 0.48 | 0.5 |
| 2.5 | 2-uphill-0 | 0.907 | 0.11 | 0 | 0 |
| 2.6 | 2-downhill-5.1 | 0.907 | 0.11 | 0.52 | 0.47 |
| 2.7 | 2-uphill-0 | 0.907 | 0.18 | 0 | 0 |
| 2.8 | 2-downhill-5.5 | 0.907 | 0.18 | 0.52 | 0.47 |

### 3.2. Winching Operations

Winch operations represent a finite share in the total time and fuel consumption as well, and they consist of the following three intervals: waiting in neutral $t_n$, winching logs $t_w$, load discharge and pulling in case of slippage $t_{add}$. Terrain slope and pulling distances are random, and set between 5 and 35 m in distance and for slopes between 10 and 30 degrees. According to the literature [28], the average winching time for a medium-large skidder is about 2 min, while for low powered farm tractors also used in skidding applications [29], this winching time may increase up to several minutes. In this paper, it is assumed that during winching the skidder is in neutral gear while workers wrap winch cable around the log and pull it towards the protection board. This time is calculated according to the following expression, proportional to log mass and winching distance:

$$t_n = 3\times + 2.5 \times m_t, \tag{5}$$

where $l_w$ is the winching distance and $m_t$ is the log mass. Coefficient values in Expression (5) are defined heuristically to obtain different waiting times proportional to mass and distance that are still close to the expected ones. Winching speed $v_w$ is set at 1 m/s and will be constant throughout this work. Total fuel consumption while the skidder is stationary and in neutral gear $g_n$ is calculated by using the following expression:

$$g_n = t_n \times g_i \tag{6}$$

where $g_i$ is fuel used while idling (set to 2.27 l/h according to [30]). Actual on-site video recorded driving scenarios [31] indicate a strong possibility of slippage happening, so it is chosen that this phenomenon appears in every other driving mission (once in two driving missions), with pulling distance set at 10 m, slope at 20° and average load mass and pulling factor ($m_{t,av} = 4.025$ t, $\mu_{p,av} = 0.49$). One complete driving cycle consists of driving to the felling site, winch operations and pulling logs to the landing site:

$$T_{cycle} = 2t_{drive} + t_w + t_n + \frac{1}{2}t_{add}, \tag{7}$$

where $T_{cycle}$ is time taken for one driving cycle, $t_w$ winching time, $t_n$ waiting time, $t_{drive}$ driving time and extra time added for slippage $t_{add}$. Fuel consumed for one driving cycle is calculated according to the following expression:

$$g_{cycle} = 2g_{drive} + g_w + g_n + \frac{1}{2}g_{add}, \tag{8}$$

where $g_{cycle}$ is total fuel consumed, $g_{drive}$ fuel consumed for driving, $g_w$ for winching, $g_n$ for idling and $g_{add}$ extra fuel added for slippage (once in two driving missions).

The parameters for eight winching operations are shown in Table 6. Logs' masses are the same as those used in Table 5.

**Table 6.** Winching operations.

| Winching Operation | Winching Length $l_w$ [m] | Log Mass $m_t$ [t] | Slope $\alpha$ [°] | Pulling Factor $\mu_p$ |
|:---:|:---:|:---:|:---:|:---:|
| 1 | 10 | 4.5 | 14 | 0.6 |
| 2 | 23 | 6 | 25 | 0.56 |
| 3 | 9 | 1.3 | 19 | 0.34 |
| 4 | 35 | 3.6 | 21 | 0.51 |
| 5 | 21 | 3 | 11 | 0.43 |
| 6 | 15 | 6 | 27 | 0.42 |
| 7 | 19 | 5.1 | 16 | 0.6 |
| 8 | 31 | 5.5 | 20 | 0.48 |

### 3.3. Reference Velocity of Custom Driving Cycle

Reference velocity is predefined depending on route slope, terrain parameters and load. Total traction force $F_l$ given by Equation (1) is multiplied by the effective wheel radius $r_w$ in order to obtain the driveline load torque $T_l$, which is then compared to the maximum driveline torque of every gear ratio (cf. Figure 1b) in order to find a feasible gear ratio. The maximum gear ratio that satisfies the feasibility condition is chosen as the gear ratio used in simulations. In order to obtain more realistic driving scenarios, the gear limit, considering negative terrain slope, is also selected. In case of slopes with less than −22°, the maximum gear is limited to the third gear, and for slopes less than −18°, the fourth gear is set as the maximum gear. Figure 10 shows the map of corresponding gear ratios considering load weight and terrain slope obtained by the above methodology.

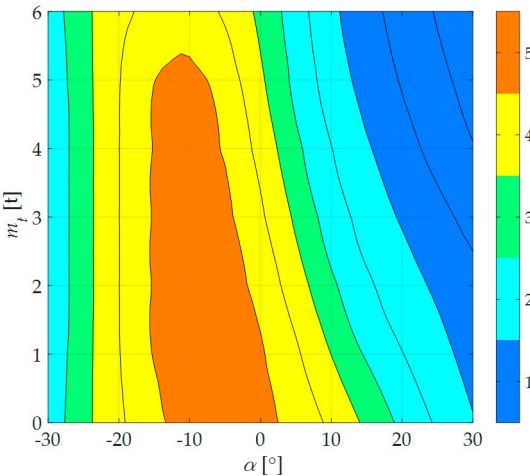

**Figure 10.** Gear depending on mass and slope.

For each gear determined by the logic described above, the reference velocity value is obtained and listed in Table 7. The reference speed for the fourth and the fifth gear are set to values that correspond to 1600 min$^{-1}$ of engine rotational speed, while for the first, second and third gears, these values tend to be above 2000 min$^{-1}$. In that way, a wider range of engine operating regimes is covered, for both large and lower loads. A problem may appear with sudden jumps of reference velocity caused by the polygonal model of the terrain profile (Figure 9). In order to avoid sudden accelerations and breaking, a Savitzky–Golay filter [32] is used to smooth out the reference velocity. Raw and filtered reference velocities of a custom driving cycle are shown in Figure 11.

**Table 7.** Reference speed values.

| Gear | Reference Velocity [km/h] |
|------|---------------------------|
| 1 | 2.8 |
| 2 | 4.7 |
| 3 | 7.9 |
| 4 | 11.2 |
| 5 | 15.5 |

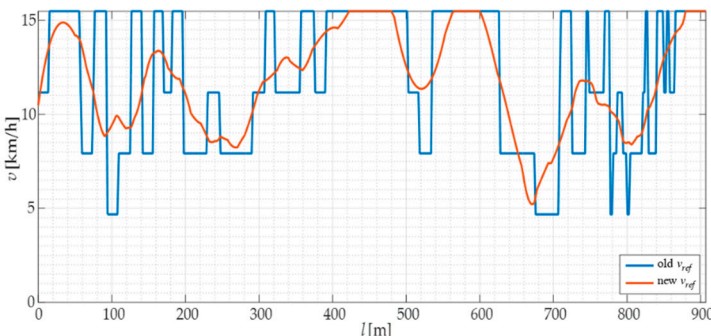

**Figure 11.** Reference velocity before and after filtering with a Savitzky–Golay filter.

## 4. Simulation Results

The main variables used in analysis are fuel consumption and time spent skidding. For each work day of the skidder, eight different sets of parameters for road and load are used, which are listed in Table 5. Time diagrams for downhill driving for driving scenario 2.2 are shown in Figures 12 and 13.

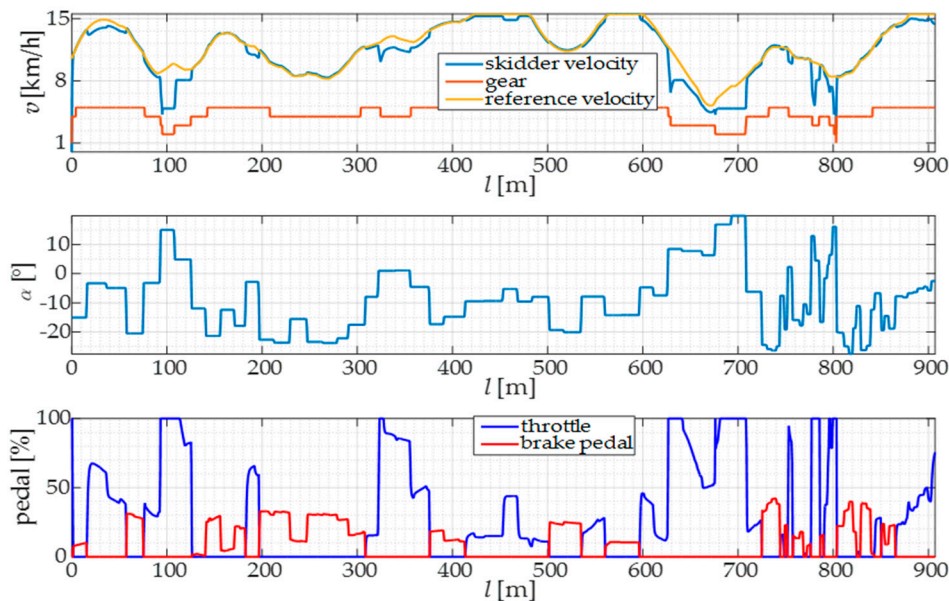

**Figure 12.** Velocities and slope for one driving scenario and driver commands.

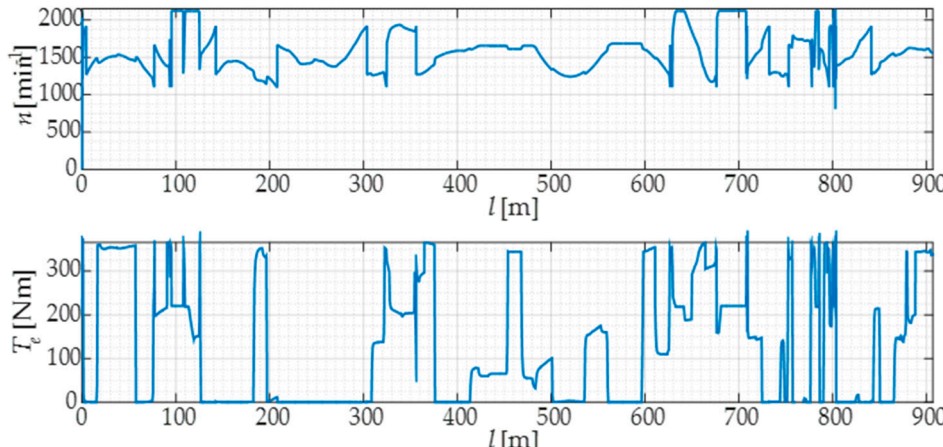

**Figure 13.** Engine speed and torque of conventional skidder for one driving scenario.

Time diagrams in Figure 12 show the skidder's velocity and gear ratio, terrain slope and driver's commands: throttle and brake pedal. They are the same for both the conventional and the hybrid vehicle.

Figure 13 shows time diagrams with engine parameters: engine speed *rpm* and engine torque for the case of the conventional vehicle. Note that the majority of driving is taking place downhill, so that the largest braking efforts are recorded at negative inclines, with the maximum negative slope amounting to—25 degrees.

Time diagrams in Figure 14 show the comparative engine torque traces for the conventional and hybrid vehicle, and the hybrid drive electromotor torque (used for ICE torque assisting and regenerative braking). A notable difference in fuel consumption can also be seen in Figure 14. A change in battery SoC is seen in the third time diagram, and is increasing when regenerative braking occurs and also in those situations when the diesel engine gives more torque than needed for driving, which is then used to recharge the battery.

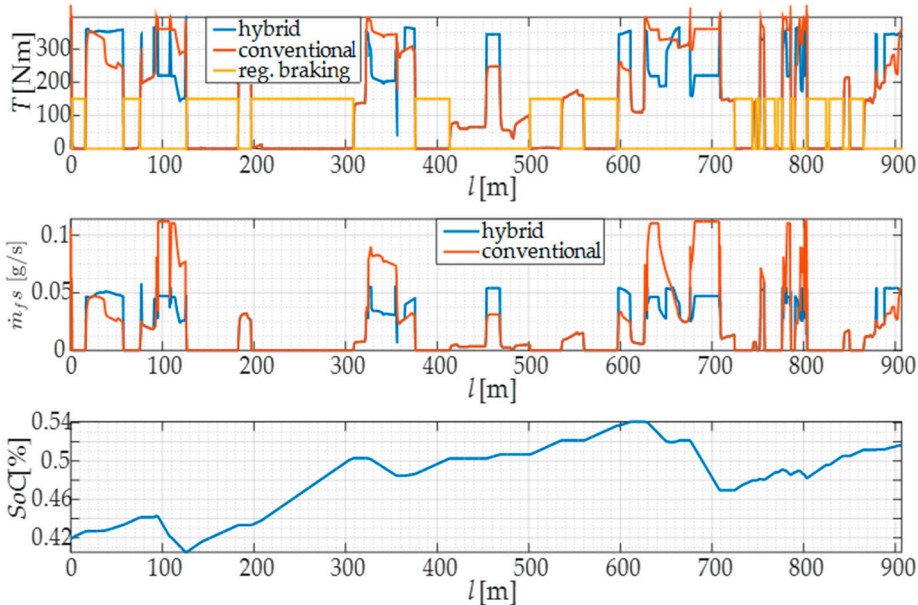

**Figure 14.** Time diagrams of hybrid vs. conventional vehicles.

Overall results for the skidder's work day are shown in Table 8.

**Table 8.** Overview of driving scenarios for the skidder's work day.

| Driving Scenario | Work Cycle Time $t$ [s] | Conventional Skidder Fuel Consumption $g_c$ [L] | Initial SoC vs. SoC Change $SoC_0/\Delta SoC$ [%] | Hybrid Skidder Fuel Consumption $g_h$ [L] |
|---|---|---|---|---|
| 1.1 | 455 | 1.75 | 65/−23 | 1.46 |
| 1.2 | 450 | 1.22 | 41/+10 | 1.12 |
| 1.3 | 436 | 1.64 | 51/−11 | 1.49 |
| 1.4 | 556 | 1.96 | 36/+4 | 1.82 |
| 1.5 | 408 | 1.52 | 40/+2 | 1.47 |
| 1.6 | 359 | 0.74 | 41/+14 | 0.67 |
| 1.7 | 385 | 1.37 | 55/−8 | 1.24 |
| 1.8 | 376 | 0.80 | 43/+16 | 0.71 |
| 2.1 | 408 | 1.50 | 59/−15 | 1.31 |
| 2.2 | 413 | 1.06 | 42/+10 | 0.95 |
| 2.3 | 397 | 1.43 | 52/−10 | 1.27 |
| 2.4 | 478 | 1.32 | 40/+11 | 1.16 |
| 2.5 | 382 | 1.35 | 51/−10 | 1.20 |
| 2.6 | 451 | 1.28 | 37/+17 | 1.06 |
| 2.7 | 446 | 1.63 | 54/−12 | 1.49 |
| 2.8 | 516 | 1.50 | 36/+13 | 1.39 |

These results show that load mass $m_t$ has the greatest impact on fuel consumption and total skidding time. For example, between driving scenarios 4 and 6, fuel consumption is larger by 1.22 liters and skidding time is longer by 197 s. Large differences also appear in driving scenarios without load. For different rolling factors, driving times vary up to 70 s and fuel consumption up to 0.28 L (driving Scenarios 1 and 7).

In all simulations, the hybrid skidder starts every work day with 65% SoC. In every driving scenario, fuel savings of between 5 and 13% are achieved. Battery SoC is increased in every instance of downhill driving considered in this work owing to the regenerative braking properties of the hybrid drive. Battery SoC increase is also greater when log mass is smaller. At the beginning of every drive with load, battery SoC is decreased by a certain percentage during winch work, as shown by the data in Table 8. After eight driving cycles, the battery SoC is at 49%, which is a drop of only 16% with respect to the initial state, so it can be said that the battery SoC is sustainable throughout the work day.

Total time, fuel consumption and change in SoC for winch operations, are shown in Table 9.

**Table 9.** Winching operations and their characteristics.

| Winching Operation | Neutral vs. Winching Time $t_n/t_w$ [s] | Winching vs. Neutral Fuel Consumption $g_w/g_n$ [L] | State-of-Charge Variation ΔSoC [%] |
|---|---|---|---|
| 1 | 41/10 | 0.03/0.26 | −1.8 |
| 2 | 84/22 | 0.73/0.053 | −4.8 |
| 3 | 30/9 | 0.066/0.019 | −0.3 |
| 4 | 114/34 | 0.072/0.072 | −4.2 |
| 5 | 71/20 | 0.039/0.044 | −1.8 |
| 6 | 60/15 | 0.041/0.038 | −2.3 |
| 7 | 70/19 | 0.059/0.044 | −3.9 |
| 8 | 107/30 | 0.088/0.067 | −5.3 |

From Table 9, it can be concluded that waiting time can be several times longer then time needed for winching and is also characterized by greater fuel consumption (operation 5). Starting SoC for the hybrid skidder is set to 65% for every scenario. During winching, battery SoC decreases up to 4.8% (operation 2).

Cycle time for one work day varies between 914 and 1103 s, with an average time of 960 s. Fuel consumption for conventional vehicle varies between 2.33 and 3.75 liters, with an average consumption of 2.88 liters, and for hybrid between 1.95 and 3.31 with average consumption of 2.49 liters, which represents savings of 13.5%. For an eight-hour work day, the skidder can perform 30 skidding cycles, which results in the total fuel consumption of 86.3 liters per single eight-hour shift. Results for one work day for the conventional and hybrid skidder are shown in Table 10. Non-productive time is not considered in this paper and results are given for the theoretical maximum, i.e., the eight-hour shift.

**Table 10.** Results for skidder's work day.

| Cycle | Cycle Time $T_{cycle}$ [s] | Conventional Skidder Cycle Fuel Consumption $g_{c, cycle}$ [L] | Hybrid Skidder Cycle Fuel Consumption $g_{h, cycle}$ [L] |
|---|---|---|---|
| 1 | 961 | 3.04 | 2.67 |
| 2 | 1103 | 3.75 | 3.31 |
| 3 | 806 | 2.36 | 2.14 |
| 4 | 914 | 2.33 | 1.95 |
| 5 | 917 | 2.64 | 2.26 |
| 6 | 955 | 2.84 | 2.43 |
| 7 | 922 | 2.76 | 2.26 |
| 8 | 1099 | 3.30 | 2.88 |
| Average | 960 | 2.88 | 2.49 |

## 5. Overall Results Discussion

Total results for conventional skidder's work days and shares of work parts in fuel consumption and time are shown in Table 11.

**Table 11.** Overall results for skidder's work day.

| Work Day | Time | Share in Time [%] | Fuel Used [L] | Share in Consumption [%] |
|---|---|---|---|---|
| Total | 8 h | | 86.3 | |
| Driving | 7 h 12 min | 90 | 82.76 | 96 |
| Winch | 12 min | 2.5 | 2.12 | 2.46 |
| Idling | 36 min | 7.5 | 1.41 | 1.54 |

There are 285 work days in one year, and assuming that the skidder works eight hours every day, by using the proposed hybrid drive configuration it is possible to save 2724 liters of diesel fuel every year, which also corresponds to 6.54 t of $CO_2$. With a battery price of 159 euros per kWh of battery [33], an electromotor price of about 5000 euros [34], and a current diesel price according to [35], it is possible to pay off the main hybrid drive components in 24 months of continuous work. The aforementioned skidder part payoff within 24 months represents a theoretical maximum, which is achievable with full work-shifts every work day. In real life scenarios, working days strongly depend on weather conditions, exploitation strategy, and similar. According to data presented in [14,36,37], the average number of operating hours in one year is 1164, which modifies the original calculations, resulting in additional parts payoff within 46 months of continuous work and the associated $CO_2$ reduction of 3.34 t per year for the hybrid powertrain-based skidder. According to [14], there are 121 EcoTrac 120 V skidders in Croatia, so it is expected that by introducing the proposed hybrid system within a greater number of vehicles, bulk purchase price of hybrid components would be lower [34], and the overall fuel and $CO_2$ savings indices would be improved.

## 6. Conclusions

The paper has presented a quasi-static model of a skidder forestry vehicle based on the available field data. The data that has been missing or incomplete has been specified based on data from vehicles with similar operating characteristics. By using two different driving routes arranged in relation to the skidder's work days, driving simulations have been carried out with different road and load parameters for the purpose of calculating the vehicle fuel consumption. When selecting the speed reference, load mass and lengths of routes, special attention has been given to matching realistic operating conditions.

After carrying out all of the required simulations for the conventional power-train vehicle, vehicle power-train hybridization is carried out. By analyzing the conventional power-train simulation results, along with work conditions and operating modes, P2 parallel hybrid configuration has been selected as the most suitable one for this application. When selecting the hybrid power-train configuration, emphasis is given to minimum changes being made to the existing vehicle power-train and chassis. By putting the EM between the diesel engine and gearbox, and separating them by two independent clutches, parallel drive configuration between the ICE and EM is established. Specifically, three operating modes are possible: (i) diesel engine only, (ii) diesel engine supported by EM, and (iii) diesel engine as the driving motor also used for parallel battery recharging. In the proposed configuration, EM completely takes over the driving of the hydraulic pump, winch drive and other hydraulic actuators. The battery and EM are dimensioned and parameterized according to the known winch characteristics, with the goal of completely taking over its power requirement from the hydraulic system. The control strategy is determined according to the fuel consumption map. In the case of near-empty battery and hybrid drive operating in the vicinity of the optimal operating point, the diesel engine is used to recharge the battery. EM is primarily used to aid the diesel engine during high-load intervals, and, in doing so, discharges the battery.

All simulations of the hybrid drive are made with the same load and route parameters as in the case of the conventional vehicle, and the respective fuel consumption results of the conventional and hybrid skidder are subsequently compared. Total savings in fuel together with the price of hybrid power-train components are shown, and they indicate that fuel savings of around 13.5% can be achieved with the proposed hybrid configuration and control strategy.

Since this paper presents only the preliminary analysis based on the available data recorded during field tests, the next step would be to compare these results with the results obtained by using other, more sophisticated control strategies. Although the proposed hybrid power-train configuration does not mandate significant interventions into the existing vehicle set-up, power-train selection in the actual vehicle will heavily depend on the free mounting space, construction change possibilities and costs of those actions.

**Author Contributions:** Conceptualization, J.K. and M.C.; methodology, J.K., J.B., M.C. and J.K.; validation, M.C., D.P., Z.Š., Z.P. and M.Š.; writing—original draft preparation, J.K.; writing—review and editing, M.C., D.P., Z.P., and M.Š.; visualization, J.K. and M.C.; supervision, Ž.Š. and M.Š.; project administration, M.Š. All authors have read and agreed to the published version of the manuscript.

**Funding:** It is gratefully acknowledged that this research has been supported by the EU European Regional Development Fund under the grant KK.01.1.1.04.0010 (HiSkid).

**Acknowledgments:** We would like to pay our gratitude and our respects to our colleague, Joško Petrić. After helping to initiate this research, Joško Petrić passed away in July of 2019. He was a tenured professor at the Department of Robotics and Production System Automation at the Faculty of Mechanical Engineering and Naval Architecture, University of Zagreb, Croatia.

**Conflicts of Interest:** The authors declare no conflict of interest.

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
