# Peer review of "Simulation Models of Skidder Conventional and Hybrid Drive"

_forests, doi:10.3390/f11090921_

Round 1
Reviewer 1 Report
Line 31: …significant fuel savings..: statistically significant? Please be more specific
Line 72: chapter 2 and after: please make clear distinction what is methodological part in what are results. Chapter Research methods should be separated.
Line 80: please add displacement of the engine (cc)
Line 123: please add units (kg, o, …)
Line 138: please be more specific what means “shorter routes”. Do you mean skidding distances? Specify the length.
Line 144: acceptance place. Do you mean landing area?
Line 212 and 213: please explain, especially 220 m/s
Line 214: chapter 3 Results should be divided into subchapters like 3.1 Results for conventional vehicle, 3.2 Results for…, etc.
Line 217: route 3? Weren’t there 2 routes (skidding trails) simulated?
Line 218: please add legend
Line 229 and 230: should be in chapter Research methods, please specify sets of load mass and terrain parameters
Line 231: please add legend
Line 233 to 237: unclear: where is load mass 1 t in Table 4? Please mark mentioned results (which route) also in Table 4 to make the conclusions clear.
Line 235 and 236: … different rolling factors that represent terrain status (mud, dry, wet) … is that mentioned in chapter Research methods?
Line 238: does it match with line 215? Please specify “significant share”.
Line 238 to 241: please consider to be a part of the Research method chapter
Line 243 and 249: please consider to be a part of the Research method chapter
Line 251: please add a legend
Line 258 to 265: please consider to be a part of the Research method chapter
Line 266 to 269: did you consider any non-productive time?
Line 268: 30 cycles per day is extremely high efficiency, is it to be real on skidding distances up to 1 km?
Line 275: time structure is not realistic, see comment lines 266 to 269. In this case fuel consumption and fuel saving is larger as in real time structure.
Line 277: chapter 4 as a whole: please consider some parts to be moved to chapters Introduction/ Research methods
Line 396, Table 11: please see comment in Line 217 and 218
Line 402 to 404: final results and especially comparison of classical and hybrid version should be more detailed.
Line 405: chapter Conclusions: unambiguous and clear answers to hypothesis should be presented in this chapter as well as comparison to results of other authors (if available – if not it should be also stated). Please consider what fuel saving means for the company and environment (more than 100 units in use).
Author Response
We would like to thank the reviewer for his/her effort in reviewing our paper and taking notice of the key aspects of our work. We hope that the revisions made per reviewer’s remarks, suggestions and requests, as well as those requests made by other reviewers will further improve the quality of our contribution. Please note that, we carefully revised our paper according to the comments and all modifications are noted by using the "Track Changes" function present in the attached manuscript.

Reviewer 2 Report
How were the load of the log transferred to the skidder - one considered a flexible log that creates an area load and the second is rigid log with a point load. It is not clear what assumption was used in the model.
The paper needs a graph that shows side by side velocity and fuel consumption for the two sliders.
The paper could use further editing. a few that I found are:
line 75 - used in different forestry estates. should be changed to used by forestry estates.
line 160 - A feasible driving cycles is .... sentence is not clear - are these the components of the skid trail - but there is no soil strength such as penetration index for soil strength.
Line 305 - completely take over - change to control.
Line 348 in every time. reword the sentence as this is confusing.
Line 392 and is bigger the load is smaller... rewrite the
Does the strategy change for skidder - smaller loads and more trips or fewer trips and larger loads? can you answer this in the discussion?
Author Response

(The authors gave the same response as above.)

Round 2
Reviewer 1 Report
Some suggestions were taken into consideration, some not.
Changes have also been done in Table 11 - but it would be very difficult (actually impossible) in real average working day to get time structure as indicated. Also 285 work days per year is very optimistic for forestry operations (rain, snow, …). Theoretically possible, in real life, I'm afraid, not.
Author Response
We would like to thank the reviewer for his/her effort in reviewing our paper and taking notice of the key aspects of our work. We agree that it would be very difficult for skidders to operate almost eight hours each day. However, we assumed that would be a theoretical maximum of skidders’ useful operation which we explained in the following sentence added in the paper:
Simulations are based on the productive time spent winching and skidding, which means that skidders work an ideal full shift every work day, without any breaks between operations. Both conventional and hybrid skidder will have the same working times.
Later, in section Overall results discussion, we compare the aforementioned results with more realistic working times from references [14,36,37]. In order to reflect this, the following text is added at the appropriate place in the text:
The aforementioned skidder part payoff within 24 months represents a theoretical maximum, which is achievable with full work-shifts every work day. In real life scenarios working days strongly depend on weather conditions, exploitation strategy, and similar. According to data presented in [14,36,37], the average number of operating hours in one year is 1164, which modifies the original calculations, resulting in additional parts payoff within 46 months of continuous work and the associated CO2 reduction of 3.34 tons per year for the hybrid powertrain-based skidder.
We hope that the revisions made per reviewer’s remarks, suggestions and requests, will further improve the quality of our contribution. Please note that, we carefully revised our paper according to the comments and all modifications are noted by using the "Track Changes" function present in the attached manuscript.
